# The Influence of an Adrenergic Antagonist Guanethidine on the Distribution Pattern and Chemical Coding of Caudal Mesenteric Ganglion Perikarya and Their Axons Supplying the Porcine Bladder

**DOI:** 10.3390/ijms22094896

**Published:** 2021-05-05

**Authors:** Agnieszka Bossowska, Ewa Lepiarczyk, Paweł Janikiewicz, Barbara Wasilewska, Urszula Mazur, Włodzimierz Markiewicz, Mariusz Majewski

**Affiliations:** 1Department of Human Physiology and Pathophysiology, School of Medicine, Collegium Medicum, University of Warmia and Mazury in Olsztyn, Warszawska 30, 10-082 Olsztyn, Poland; ewa.lepiarczyk@uwm.edu.pl (E.L.); pablo3991@gmail.com (P.J.); bachaw@uwm.edu.pl (B.W.); urszula.mazur@uwm.edu.pl (U.M.); mariusz.majewski@uwm.edu.pl (M.M.); 2Department of Pharmacology and Toxicology, Faculty of Veterinary Medicine, University of Warmia and Mazury in Olsztyn, Oczapowskiego 9, 10-719 Olsztyn, Poland; mark@uwm.edu.pl

**Keywords:** guanethidine, noradrenergic nerve fibers, inferior mesenteric ganglion, neuropeptides, urinary bladder, pig

## Abstract

This study was aimed at disclosing the influence of intravesically instilled guanethidine (GUA) on the distribution, relative frequency and chemical coding of both the urinary bladder intramural sympathetic nerve fibers and their parent cell bodies in the caudal mesenteric ganglion (CaMG) in juvenile female pigs. GUA instillation led to a profound decrease in the number of perivascular nerve terminals. Furthermore, the chemical profile of the perivascular innervation within the treated bladder also distinctly changed, as most of axons became somatostatin-immunoreactive (SOM-IR), while in the control animals they were found to be neuropeptide Y (NPY)-positive. Intravesical treatment with GUA led not only to a significant decrease in the number of bladder-projecting tyrosine hydroxylase (TH) CaMG somata (94.3 ± 1.8% vs. 73.3 ± 1.4%; control vs. GUA-treated pigs), but simultaneously resulted in the rearrangement of their co-transmitters repertoire, causing a distinct decrease in the number of TH^+^/NPY^+^ (89.6 ± 0.7% vs. 27.8 ± 0.9%) cell bodies and an increase in the number of SOM-(3.6 ± 0.4% vs. 68.7 ± 1.9%), calbindin-(CB; 2.06 ± 0.2% vs. 9.1 ± 1.2%) or galanin-containing (GAL; 1.6 ± 0.3% vs. 28.2 ± 1.3%) somata. The present study provides evidence that GUA significantly modifies the sympathetic innervation of the porcine urinary bladder wall, and thus may be considered a potential tool for studying the plasticity of this subdivision of the bladder innervation.

## 1. Introduction

The urinary bladder has two main correlated functions: storage of urine and its periodical and voluntary voidance. These abovementioned functions are regulated by a complex neural control system containing somatic, sympathetic, and parasympathetic components. Peripheral nerves coordinate the activities of a variety of effector structures including the smooth muscle of the urinary bladder and the smooth and striated muscle of the urethral sphincters [1,2] and their function is modulated by higher nervous centers, including the cerebral cortex, the cerebellum and pons [3].

Undisturbed transmission in the autonomic nerves supplying the urinary bladder wall is essential for the proper urine storage processes [4]. The urinary bladder detrusor muscle has double, acting contradictory, sympathetic-parasympathetic innervation. These two neuronal pathways are a structural basis for reflexes which either keep the bladder in a relaxed state, enabling urine storage at low intravesical pressure, or initiate bladder emptying by relaxing the outflow region and contracting the detrusor muscle [5]. The parasympathetic innervation of the urinary bladder is carried out by pelvic splanchnic nerves coming from the intermediomedial nucleus of the sacral segments of the spinal cord [6] and consists mainly of preganglionic fibers supplying the intramural ganglia, which in turn send the postganglionic fibers [7,8]. The main function of the parasympathetic innervation of this organ is triggered by the acetylcholine (ACh) released from the parasympathetic postganglionic neurons. This neurotransmitter activates muscarinic receptors in the bladder [9] and therefore mediates contraction of the detrusor muscle and urination [10]. The sympathetic pathway to the urinary bladder begins in the intermediolateral nuclei of the thoracic (Th) and lumbar (L) segments of the spinal cord at the level Th10–L2 in humans and L1–L5 in dogs and cats [6]. The majority of preganglionic axons synapse then on the postganglionic neurons localized either within paravertebral (lumbosacral sympathetic chain) ganglia (SChG), or in the prevertebral ganglia, including the inferior mesenteric ganglion (IMG; in animals this ganglion is called caudal mesenteric ganglion—CaMG) [11,12,13]. However, some of the lumbar sympathetic preganglionic axons, controlling the lower urinary tract, pass/send signal to the extramural ganglia of the pelvic plexus or to the intramural ganglia of the urinary bladder wall [14,15]. They reach the above mentioned ganglia via hypogastric nerve branches [16] and release transmitters such as noradrenaline (NA) [17,18], neuropeptide Y (NPY) [19], somatostatin (SOM) [20] and galanin (GAL) [21]. The main function of the sympathetic innervation of the urinary bladder is triggered by NA released from the sympathetic postganglionic neurons. It inhibits the bladder detrusor muscle (via β_3_-adrenoreceptors) and excites the bladder base and urethra (via α_1_-adrenoreceptors), thus achieving continence [1].

Guanethidine (GUA) is an adrenergic antagonist which belongs to organic compounds acquired from the series of guanidine derivatives [22]. For the first time it was described in 1959 by Maxwell et al. [23] as an antihypertensive, as well as a sympatholytic agent that acts by inhibiting selective transmission in postganglionic adrenergic nerves [24]. Although the exact mechanism(s) of GUA action is not fully established yet, it is well-known that GUA is neither able to cross the blood–brain, nor the blood–cerebrospinal fluid barrier and therefore its effect is limited to peripheral noradrenergic neurons [25]. Boullin [26] found that GUA acts as a “false” transmitter in rats, suggesting that it functions at the sympathetic neuroeffector junction mainly by preventing the release of NA and causing depletion of NA in the peripheral sympathetic nerve terminals, rather than by acting at the effector cell by inhibiting the association of NA with its receptors. GUA is transported across the sympathetic nerve membrane by a special mechanism using a norepinephrine transporter (NET) [26]. GUA accumulates in the synaptic vesicles, thus replacing NA, which leads to a gradual depletion of NA stores in the nerve endings. Once inside the terminal, GUA blocks the release of NA in response to the arrival of an action potential, leading to the functional sympathetic denervation [27]. However, Maxwell [28] suggested a different mechanism of GUA action: the adrenergic GUA-related block preceded by a short-term sympathomimetic effect, which is caused by a transient releasing of NA from adrenergic nerve endings leading to depolarization and stabilization of neuronal membrane. This effect of GUA is only achieved when the higher doses are administrated and it is synonymous with destruction of adrenergic neurons [29].

In the case of the urinary bladder, GUA was mainly used to block the effects of sympathetic innervation in functional tests concerning the influence of various biologically active substances and drugs on the contractile activity of this organ. These studies were carried out in humans [30] and in various animal species such as rabbits, cats, guinea pigs, sheep, rats, mice [31,32,33,34,35,36] and pigs [37,38,39]. However, based on the available literature, there are no relevant data on the influence of GUA on the neuroarchitecture of sympathetic pathways supplying the urinary bladder. Therefore, the purpose of the present study was to investigate, using combined retrograde tracing and immunohistochemistry, the putative influence of intravesical instillation of GUA on the distribution and chemical coding of the bladder sympathetic nerve fibers, as well as the CaMG neurons projecting to the urinary bladder wall. We have decided to use the domestic pig in this experiment since these animals, as opposed to smaller animals, such as cats and rats, present the advantage that the anatomy, histology and functionality of their urinary bladder resembles the human one. Therefore, pigs are considered to be an optimal species for preclinical experimentation as they allow for human-related validation of valuable research information gained during the study [40,41].

## 2. Results

### 2.1. Distribution and Relative Frequency of Dopamine Beta Hydroxylase (DβH)-Immunoreactive (IR) Nerve Fibers in the Urinary Bladder Wall of Control and GUA-Treated Animals

In the control group, a relatively small number of DβH-IR nerve fibers was observed in the smooth muscle layer of the urinary bladder wall (Figure 1a,b). A moderate number of DβH-IR terminals was observed in the submucosal layer (Figure 1d,e) and only single fibers penetrated beneath the urothelium (Figure 1g,h). However, the blood vessels observed both in the muscle coat and submucosa were densely supplied with numerous axons containing DβH (Figure 1j,k).

The GUA-treatment resulted in the distinct decrease in the number of noradrenergic axons around the blood vessels (Figure 1l) while non-vascular DβH-positive nerve fibers were virtually not observed in the urinary bladder wall (Figure 1c,f,i comprise representative images of the muscle layer, submucosal layer, and the urothelium, respectively).

Details concerning the relative density of DβH-IR nerve fibers in the particular layers of the urinary bladder wall in both the control and GUA-treated animals are shown in Table 1.

### 2.2. Colocalization Pattern of DβH and Other Substances Studied in the Nerve Fibers of the Urinary Bladder Wall in Control and GUA-Treated Animals

The double-immunohistochemical staining technique has revealed that the majority of noradrenergic axons supplying the muscle layer of the urinary bladder wall contained NPY (Figure 2a), a moderate number of DβH-IR axons stained for SOM (Figure 2b), while only single nerve terminals were vasoactive intestinal polypeptide (VIP)-, GAL- or calbindin (CB)-IR (Figure 2c,g,h, respectively).

In the submucosal layer and beneath the urothelium, only a few noradrenergic nerve fibers were simultaneously immunopositive to NPY (Figure 2i) or SOM (Figure 2n). Many DβH-IR axons observed around blood vessels were stained for NPY (Figure 2m) but only single noradrenergic nerve fibers expressed immunoreactivity to SOM (Figure 2n). DβH-IR axons that simultaneously stained for neuronal nitric oxide synthase (nNOS) were not observed in any layer of the urinary bladder wall nor around the blood vessels (Figure 2o).

After GUA treatment, chemical coding of noradrenergic axons within the muscle layer, submucosal layer, beneath the urothelium, as well as around arterial blood vessels, drastically changed. DβH-IR terminals containing either NPY (Figure 2d), SOM (Figure 2e), GAL (Figure 2j), CB (Figure 2k) or VIP (Figure 2f) were not observed within the muscle layer. Furthermore, after GUA instillation no DβH-IR nerve endings stained for NPY (Figure 2l) or SOM were found in the submucosal or suburothelial layers; however, nearly all DβH-IR perivascular axons were simultaneously SOM-IR (Figure 2r), while only single nerve terminals were immunopositive to NPY (Figure 2p). Additionally, single noradrenergic axons containing nNOS were observed in the wall of blood vessels (Figure 2s).

Details concerning the degree of colocalization of DβH and co-transmitters studied within the nerve fibers in the particular layers of the urinary bladder wall in both the control and GUA-treated animals are shown in Table 2.

### 2.3. Distribution of Fast Blue-Positive (FB^+^) Neurons in the CaMG of Control Pigs

Three weeks after the administration of fast blue (FB) into the right side of the urinary bladder wall, FB-containing sympathetic neurons were found bilaterally in CaMGs of all animals studied. A distinct “lateralization” of the urinary bladder-projecting CaMG neurons localization was observed in terms of their ipsi- and contralateral distribution, with a distinct predominance of FB^+^ nerve cells in the ipsilateral ganglia. The number of FB^+^ neurons per animal ranged from 528 to 697 retrograde labeled perikarya (616 ± 48.9; mean ± SEM) in the right ganglia and from 29 to 34 FB-positive neurons (31.3 ± 1.4) in the left ones. Approximately 89% of all FB^+^ CaMG neurons (88.9 ± 0.8%) were located in the ipsilateral ganglia, while only 11% (11.1 ± 0.8%) were found in contralateral CaMGs. With regard to their diameter, two distinct subpopulations of traced cells were distinguished in the control CaMGs: the first one consisted of large neurons (40–60 μm in diameter), constituting approximately one-fourth of all traced cells in both the ipsi- and contralateral ganglia (26.9 ± 2% and 28.2 ± 2%, respectively) and a second much more numerous subset comprised of smaller nerve cells (15–35 μm in diameter), constituting 73.1 ± 2% and 71.8 ± 2% of bladder-projecting cells, respectively.

Within the ipsilateral ganglia, the majority of the urinary bladder-projecting neurons (UBPN) were located caudally, along the lateral ganglionic border. Additionally, a small group of neurons was regularly found closer to the middle part of the ganglion. Although the majority of FB^+^ nerve cell bodies in the contralateral ganglia were also distributed in the caudal part of the ganglion, they were randomly dispersed and did not form any distinct, somatotopically organized clusters.

### 2.4. Distribution of FB^+^ Neurons in the CaMG of GUA-Treated Pigs

The abovementioned pattern of FB^+^ neurons distribution was also unchanged in the GUA-treated animals: in general, they were bilaterally distributed, with a clear ipsilateral prevalence. The number of FB-containing neurons per animal ranged from 562 to 663 retrogradely labelled perikarya (616.7 ± 29.4) in the right ganglia and from 26 to 34 FB-positive neurons (29.7 ± 2.3) in the left ones. Approximately 88% (87.9 ± 1%) of dye-labeled neurons were found in the right and about 12% (12.1 ± 1%) in the left CaMGs studied. In terms of cell diameters, there were 71.6 ± 2.4% of small and 28.4 ± 2.4% of large FB^+^ neurons in the right CaMGs. In the left ganglia, these values amounted to 70.3 ± 6.8% and 29.7 ± 6.8%, respectively. The number and the distribution of UBPN in the GUA-treated animals in both the ipsilateral and contralateral CaMG ganglia were similar to those observed in the control group.

### 2.5. Immunohistochemical Characteristics of FB^+^ Neurons in the CaMG of Control and GUA-Treated Animals

Under physiological conditions, both the tyrosine hydroxylase-immunoreactive (TH-IR; 94.3 ± 2%; Figure 3b,e,h,k,n,r), as well as TH-immunonegative (TH^−^; 5.7 ± 2%; Figure 3h,n,r) retrogradely labeled bladder-projecting neurons were found in both the ipsilateral and contralateral CaMGs studied.

Administration of GUA into the urinary bladder led to a significant decrease in the number of FB^+^ TH-IR neurons (Figure 4b,e,h,k,n,r), compared to the results obtained in the control group (73.3 ± 1% vs. 94.3 ± 2% of all FB^+^ neurons, respectively). Simultaneously, the number of TH-immunonegative cells (Figure 4n) distinctly increased (from 5.7 ± 2% in the control group to 26.7 ± 1% in GUA-treated animals).

#### 2.5.1. Co-Transmitters of FB^+^/TH-Positive (TH^+^) Neurons in the CaMG of Control and GUA-Treated Animals

Under physiological conditions, the vast majority of TH-IR traced cells (89.6 ± 1%) were simultaneously NPY-positive (Figure 3a–c). A small subpopulation of noradrenergic UBPN simultaneously co-expressed SOM (3.6 ± 0.4%; Figure 3d–f), CB (2.06 ± 0.2%; Figure 3g–i), VIP (1.6 ± 0.2%; Figure 3j–l) or GAL (1.6 ± 0.3%; Figure 3m–o). None of the FB^+^/TH^+^ perikarya were found to be immunopositive to nNOS (Figure 3p–s).

Treatment with GUA caused a distinct, statistically significant (*p* < 0.0001) decrease in the number of NPY^+^ noradrenergic neurons (27.8 ± 0.9%; Figure 4a–c) and a complete loss of retrogradely labeled TH^+^ nerve cells containing VIP (Figure 4d–f). On the other hand, intravesical instillation of GUA led to a significant increase in the number of FB^+^/TH^+^ neurons immunopositive to SOM (68.7 ± 1.9%; Figure 4g–i), GAL (28.2 ± 1.3%; Figure 4j–l), CB (9.1 ± 1.2%; Figure 4m–o) or nNOS (4.5 ± 0.6%; Figure 4p–s). There were no statistically significant differences in the number of individual subpopulations of noradrenergic retrogradely labeled nerve cells between right and left CaMG ganglia studied in the control group as well as in the GUA-treated animals.

Details concerning the relative frequency of individual subpopulations of retrogradely labeled TH^+^ neurons in both the control and GUA-treated animals are shown in Table 3.

#### 2.5.2. Immunohistochemical Characteristics of FB^+^/TH-Immunonegative Neurons in the CaMGs of Control and GUA-Treated Animals

Among FB^+^ non-adrenergic (TH^-^) neurons, numerous perikarya stained for NPY (38.9 ± 6.7%), while the second-most numerous fraction were the VIP-containing neurons (14.8 ± 6.2%). No FB^+^, but TH-immunonegative nerve cells were simultaneously immunoreactive to SOM, CB (Figure 3g–i), GAL (Figure 3m–o) or nNOS (Figure 3p–s).

The neurochemical characterization of TH-immunonegative CaMG neurons supplying the urinary bladder after GUA treatment was significantly different from that observed in the control animals. The changes were associated primarily with a distinct decrease in the number of TH-immunonegative UBPN stained for NPY (27.8 ± 0.9% in GUA-treated animals vs. 38.9 ± 6.7% in the control group). Furthermore, GUA administration into the urinary bladder was followed by a complete elimination of FB^+^/TH-immunonegative nerve cells simultaneously containing VIP. As in the control group, no TH-immunonegative perikarya containing SOM, CB (Figure 4m–o), GAL or nNOS were observed in the CaMGs of the GUA-treated pigs. There were no statistically significant differences in the number of individual subpopulations of non-adrenergic retrogradely labeled nerve cells between right and left CaMG ganglia studied in the control group and in the GUA-treated animals.

Details concerning the relative percentages of the individual subpopulations of retrogradely labeled TH-immunonegative neurons in both the control group and in the group of GUA-treated animals are summarized in Table 4.

## 3. Discussion

The results of the present study clearly indicate that the application of GUA is followed by profound changes in the distribution, relative frequency, and chemical coding of sympathetic nerve fibers as well as the noradrenergic CaMG neurons supplying the wall of the porcine urinary bladder. Additionally, the present study confirms that CaMG plays a significant role in the autonomic innervation of the bladder in the female domestic pig.

A thorough discussion regarding the distribution, relative frequency, and chemical coding of both noradrenergic nerve fibers and CaMG neurons supplying the wall of the female porcine urinary bladder was already presented in our previous papers [42,43]. However, it should be stressed that in both previous studies control groups were slightly different: while in the case of the study focusing on intramural noradrenergic nerve fibers [42], no medical procedures were applied to the control pigs, in studies concerning the neurochemical characteristics of the noradrenergic CaMG neurons supplying the porcine urinary bladder, the control pigs were either intravesically instilled with 5% aqueous solution of ethyl alcohol [43], or with citrate buffer (present experiment). The aim of the abovementioned procedures performed in all the control pigs was to guarantee that changes in the distribution and chemical coding of bladder noradrenergic nerve fibers and CaMG neurons observed after the treatment with toxins (botulinum toxin type A and resiniferatoxin in the previous studies) as well as the GUA (this study) were caused by these biological active substances themselves, and not due to factors associated with the technique and route of their administration. Furthermore, it should be noted that the number, age, body weight and sex of the animals used as the control groups in mentioned experiments, as well as all the surgical and immunohistochemical procedures applied, were entirely corresponding. Accordingly, no significant differences in the distribution as well as the chemical coding of noradrenergic nerve fibers and in the number of FB^+^ CaMG-UBPN or in the percentages of the immunopositive neurons were observed between the control animals of the present study compared to the control groups used in the previous experiments. Therefore, in the present discussion, we are focusing on the data concerning changes caused by GUA treatment.

The present findings strongly suggest that GUA is a factor evoking distinct adaptational changes in autonomic neurons supplying the urinary bladder wall. These changes include modifications of the chemical phenotype and/or alterations in both the density of noradrenergic nerve fibers and the number of neurons originating from CaMGs. The present study has shown that GUA distinctly decreased the number of noradrenergic nerve terminals in all the layers of the urinary bladder wall. Simultaneously, a statistically significant decrease in the number of TH^+^ CaMG neurons projecting to the urinary bladder wall was observed.

In the earlier studies, it has been shown that the intravenous injection of GUA decreases the NA content of heart muscle, intestine and spleen in rabbits and rats [44,45,46,47,48], of heart muscle and aorta in dogs, and vas deferens [49] and kidney in rats [50]. Recent studies have shown similar results in the lower urinary tract. The inhibitory effect of GUA on the noradrenergic innervation was observed in the bladder detrusor in mice [36] and rats [51], and in the urinary bladder neck in pigs [38]. It is believed that decrease in the density of noradrenergic fibers in target tissues caused by the action of GUA is related to its direct sympathomimetic effect leading to the release of NA from stores in the effector organs [52], which is followed by inhibition of the re-uptake and re-synthesis of NA, leading to its depletion in the noradrenergic varicosities [25]. In respect to the sympathetic ganglia, the loss of TH activity was found in the superior cervical ganglion in rats after chronic administration of GUA [53]. These morphological findings corroborate and explain the observations made by Loh and collaborators [54], who found that a local infusion of GUA in men prevents sympathetic activity and stops the pain, hyperpathia and allodynia, reduces nerve conductance to a minimum, and causes vasodilatation.

The action of NA on the entire urinary tract, and in particular on the urinary bladder, is complex and depends on both the type of stimulated adrenergic receptor and its location in the urinary tract. Beta receptors predominate in the bladder body, whereas alpha receptors predominate in the base [55]. The key function of β-adrenoceptors in the bladder in many species including pigs [56,57] is smooth muscle relaxation and an increase in bladder compliance during the filling phase of the micturition cycle. In case of α_1_-adrenoceptors, on the one hand, it has been found that they are functionally expressed by capsaicin-sensitive primary sensory neurons in the urinary tract [58], being deeply involved in nociceptive transmission from the bladder and in the neurogenic inflammation process. On the other hand, α_1_ adrenergic receptors are involved in the activation of the bladder mechanosensory Aδ fibers and C fibers during the bladder filling phase [59]. The sensory neuropeptides neurokinin A and calcitonin gene-related peptide (CGRP) have been found to be released by sensory fibers after stimulation by either single pulse electrical field stimulation or capsaicin, thus inducing the contraction of the urinary bladder [60]. These results suggest that the activation of α_1_ adrenergic receptors in the sensory nerves innervating the urinary bladder wall may lead to the release of some neuropeptides that modulate detrusor muscle tone. Additionally, it was postulated that α_1_ adrenergic receptors may mediate prejunctional facilitation causing the release of acetylcholine [61,62] or norepinephrine [61] from the nerve terminals. It has been shown that in most species, including humans, α_1_-adrenoceptor stimulation produces weak detrusor contraction, whereas a stronger contraction is observed in the trigone, bladder base and/or in the bladder neck [63]. On the other hand, it has been observed that activation of α_1_ adrenergic receptors in the urothelium in rats [64], and the treatment of sensory submucosal nerve fibers in mice [36] with phenylephrine leads to the release of some unknown neurotransmitters from both sources, which in turn facilitate detrusor relaxation, probably by augmentation of noradrenaline release from the sympathetic nerves. Subsequently, noradrenaline stimulates β-adrenoceptors on the detrusor muscle, leading to relaxation of the urinary bladder [36]. Another mechanism, observed in the bladder neck, relies on prejunctional activation of α_2_ adrenoceptors by NA and leads to nitric oxide (NO) release causing smooth muscle relaxation [65].

Considering the NA action on the vesical vessels, in the in vivo model of the rat urinary bladder microcirculation this compound caused a statistically significant decrease in vascular diameters of both arterioles and venules, i.e., vasoconstriction [66]. However, studies performed on rat mesenteric resistance in blood vessels have shown that activation of sympathetic nerve terminals leads to release of NA (or related substance(s)), which then activates the sensory nerves to release CGRP, causing vasodilation [67]. Considering the abovementioned data, it seems likely that the main effect of the intravesical GUA instillation may be an increase in the contractile activity of the detrusor muscle with the simultaneous relaxation of the muscles in the bladder neck, which in turn may lead to an increase in urine output. It also seems possible that GUA may affect the vascular smooth muscle leading to their relaxation and increased blood flow through the wall of the bladder. However, this hypothesis must be carefully evaluated in the future.

Several studies have demonstrated that chronic treatment of either newborn or adult rats by high doses of GUA, causes destruction of peripheral sympathetic neurons (i.e., a kind of pharmacological sympathectomy) [68,69,70]. For example, it has been shown that GUA produced a marked depletion of sympathetic neurons in the celiac/mesenteric and superior cervical ganglia [71]. However, no statistically significant differences in the number and distribution pattern of CaMG-UBPN between control and GUA-treated pigs were observed in the present experiment. This finding suggests that, at least in the case of the present study, the GUA instillation into the urinary bladder, did not evoke neuronal death.

Generally, GUA is considered to be a factor causing the progressive degeneration of sympathetic neurons. It has been suggested that a selective degenerative effect of GUA on adrenergic neurons may be caused by the presence of an amine uptake pump [72]. Several mechanisms have been proposed to account for the cytotoxic effects of GUA, including the inhibition of oxidative phosphorylation [73,74], the inhibition of the retrograde transport of nerve growth factor (NGF) [75], and/or the inhibition of polyamine biosynthesis [76]. It has been also suggested that GUA may exert its cytotoxic effects by an immunologically mediated mechanism(s); however, the precise immune process responsible for the neuronal destruction still remains to be elucidated; it may involve delayed-type hypersensitivity, cytotoxic T cells, antibody-dependent cell-mediated cytotoxicity, or any combination of the above [29,70]. Thanks to numerous studies, it was possible to conclude that degenerative changes observed in both the nerve fibers and the sympathetic neurons after GUA treatment are possibly caused by the sequential effect of GUA involving a primary step of NA depletion, followed by subsequent degeneration of nerve terminals and cell bodies, which are dose- and time-related. It has been found, that only high, repeated doses (30–60 mg/kg/day) are able to induce long-lasting damage of noradrenergic neurons, whereas small doses (5 mg/kg/day) only deplete NA stores [53,77]. Additionally, light microscopic examination and TH activities indicated that destruction of the superior cervical ganglion neurons in rats is complete by the second week of GUA treatment in 50–100 mg/kg/day doses [49]. For this reason, perhaps, the lack of sympathetic neuron degeneration in CaMG ganglia observed in our research may be attributed to the use of a single, small dose of GUA.

The results of the present study strongly suggest that GUA not only caused distinct changes in the synthesis/storage rate of NA but also profoundly influenced the expression of most of the investigated biologically active substances found to colocalize with NA in the sympathetic neurons supplying the urinary bladder. Thus, GUA treatment was followed by a dramatic decrease in the number of noradrenergic nerves expressing NPY in all the layers of the bladder wall and around the blood vessels. This finding is consistent with the significant decrease in the number of urinary bladder-supplying TH^+^/NPY^+^ somata. These results correspond well with data reported in the previous studies [71,78,79], in which chronic GUA treatment in adult rats caused depletion of NA and NPY not only in the superior cervical and celiac/mesenteric ganglia and their target-tissues, but also in the urinary bladder wall [78]. In addition, Mattiasson et al. [80] have demonstrated that chemical sympathectomy induces a complete depletion of NPY-immunoreactive perivascular fibers in the rat bladder.

Numerous studies have been conducted on determining the physiological function of NPY in sympathetic transmission. Exogenously applied NPY caused not only a potent reduction in local blood flow in a variety of organs and an increase in systemic arterial blood pressure in cat and pig [81,82], but also a reduction of forearm blood flow in humans [83]. One popular hypothesis, pertaining especially to blood vessels, is that NPY promotes “economy” of the motor transmitters adenosine triphosphate (ATP) and NA. For instance, it has been found that in the vas deferens in mice [84] and guinea pigs [85], upon release, NPY does not produce a contraction per se, but acts as a neuromodulator leading to prejunctional inhibition of NA release, whereas postjunctionally, NPY modulates the action of NA and ATP by enhancing their action on their receptors. In the lower urinary tract, NPY has been found, through activation of Y_2_-receptors, to potentiate phasic and tonic contractions in the horse intravesical ureter [86], while in rats, NPY increased the level of spontaneous activity of the bladder detrusor muscle [87]. Furthermore, it has been also postulated that in the rat urinary bladder NPY evoked an inhibition of the cholinergic contractions [88] and exerted a prejunctional inhibitory effect on the noradrenergic transmission [89].

GUA treatment of the porcine bladder was also followed by a total decrease in the number of non-vascular noradrenergic nerve fibers containing SOM in all layers of the organ wall, and on the contrary, GUA instillation caused a significant increase in the number of perivascular SOM-positive fibers and a concomitant increase in the number of retrogradely traced TH^+^ CaMG neurons co-expressing SOM.

The subpopulation of SOM/TH-IR neurons have been described earlier in porcine CaMG by Majewski and Heym [90] and according to Lacroix et al. [91], Majewski and Heym [90], and Majewski [92], it is possible to hypothesize a vasoconstrictor role for these neurons. In the detrusor muscle of the guinea pig and rabbit, SOM causes a concentration-dependent rise in the basal tone, but no phasic contraction [93]. SOM is also considered to be an anti-inflammatory and antinociceptive factor, which may be actively involved in the attenuation of enteritis [94] and cystitis [95]. It has been postulated that SOM modulates K^+^/Ca^2+^ channel activities that “sensitize” the sympathetic neurons to NGF and thus can facilitate the neuroprotection and regeneration [96], and this phenomenon may explain, at least in part, the observed increase in the number of FB^+^/SOM^+^ perikarya after intravesical GUA instillation.

Interestingly, intravesical treatment with GUA led to a total disappearance of both the intramural sympathetic VIP^+^ terminals, and the VIP-containing noradrenergic neurons in the CaMG. Both the unveiling of their function, and the reason behind their disappearance, requires further study. Although the vast majority of available reports support the view that VIP exerts an inhibitory action on neural pathways controlling micturition by relaxation of the smooth muscles (see below), it should be stressed that there are also some studies suggesting the opposite, i.e., excitatory action of this peptide on the detrusor [97,98]. Distribution patterns of VIP-containing nerves in the porcine urinary bladder (in the detrusor, around blood vessels and beneath the urothelium) suggests that it may participate in the regulation of smooth muscle activity, perhaps by acting as a local modulator of neuromuscular transmission, blood flow and epithelial activity/conveyance of sensory information [7]. The subsequent studies confirmed this hypothesis, showing that VIP produces relaxation in the urinary tract either by a direct action on the smooth muscle cells or by indirect effects mediated through NO release from autonomic intramural neurons [99]. This activity of VIP has been confirmed not only in humans [100] or pigs [99], but also in rabbits [101] and guinea pigs [102]; however, not in rats [103]. Further pharmacological studies in human and porcine lower urinary tract unraveled that VIP not only reduces basal tension and agonists-evoked contractions in the bladder body and base [100,104,105] and relaxes the bladder neck in pigs [106], but also can reduce the tension and amplitude of the spontaneous contractions of trigonal strips [104]. Last, but not least, it has also been reported that VIP relaxes porcine [107] and human ureter [100] as well as porcine urethra [108,109].

The present study has also revealed that intravesical instillation of GUA, on the one hand leads to a significant decrease in the density of noradrenergic nerve fibers containing GAL or CB within the muscle layer of the urinary bladder wall, and on the other hand GUA treatment was followed by a simultaneous distinct increase in the number of CaMG FB^+^/TH^+^ neurons immunoreactive to GAL or CB. The exact function of intramural nerve fiber and neurons immunoreactive to these two substances in correlation with NA in the urinary tract tissues is not clear. Some studies have shown that GAL can influence the activity of smooth muscles in rat bladder and modulate neural transmission in both autonomic ganglia and at neuromuscular junctions where it suppressed the cholinergic component of the response to electric field stimulation. Thus, these results have suggested an inhibitory action of GAL on neurotransmitter release in the smooth muscle tissues, and this may also pertain to the urinary bladder [110]. Additionally, the results obtained by Honda et al. [111] have shown that intrathecal administration of GAL in rats delays the onset of micturition, suggesting the inhibitory role of galaninergic system in the control of the micturition reflex. Moreover, it has been suggested that this neuropeptide not only plays very important anti-inflammatory and anti-nociceptive functions [112]; but, as clearly indicated in a number of studies, GAL should also be considered as one of the most important substances involved in neuroprotection, regeneration and survival of damaged neurons within the central [113] and peripheral nervous system under various pathological conditions. For example, an increase in the expression of GAL has been observed in DRG neurons after peripheral nerve injury [114] or during cystitis [115]. Furthermore, axotomy-related over-expression of GAL in the uterus-projecting neurons of the porcine CaMG has also been reported [116]. Thus, it seems likely that the simultaneous decrease in the number of GAL-positive fibers in the bladder wall (albeit requiring exact explanation in itself) and the increase in the number of GAL^+^ CaMG cells observed in this study reflect autonomic ganglion repair processes rather than remodeling of the bladder innervation pattern.

It is well documented that CB is an intracellular calcium-binding protein responsible for Ca^2+^ homeostasis in neurons and it is believed to exert neuroprotective effects on these cells by preventing them from large fluctuations in free intracellular Ca^2+^ and cell death [117,118,119]. Therefore, an increase in the number of CB^+^ noradrenergic CaMG-UBPN probably reflects a defensive reaction of investigated cells, challenged by GUA.

Our study has shown that GUA instillation into the urinary bladder induced an expression of nNOS immunoreactivity in single fibers of vesical blood vessel wall and in some CaMG-UBPN. The physiological function of the neuronal NO-mediated vasodilatation in the bladder has been speculated [120]. This observation is in line with functional studies showing that NO may have a role in the inhibitory neurotransmission in the pig trigone and seems to be involved in the endothelium dependent ACh-induced relaxation of pig vesical arteries [121,122]. Similar to GAL or CB, NO may also participate in the adaptive processes enabling the protection and survival of nerve cells under various pathological conditions. Vanhatalo and Soinila [123] have reported a massive induction of NADPH diaphorase activity in postganglionic neurons of the rat superior cervical ganglion following colchicine treatment, postganglionic nerve trunk ligation or ganglion culture. The overexpression of nNOS-containing sympathetic neurons in the porcine CaMG was observed after chemically induced colitis and after axotomy [124,125]. However, it should be noticed that in our study nNOS was expressed merely in single UBPN and thus the physiological relevance of GUA-induced changes concerning the expression of this substance is probably only, if anything, marginal.

The present results have also revealed that GUA instillation resulted in some changes in the chemical coding of TH-negative (non-adrenergic) CaMG UBPN. Precisely, administration of this drug was followed by a statistically significant decrease in the number of FB^+^/TH^-^ neurons expressing immunoreactivity to NPY or VIP. These results strongly indicate that GUA not only affects adrenergic neurons, but also can modify the function of non-adrenergic (thus possibly cholinergic or non-adrenergic, non-cholinergic) nerve cells in CaMGs.

## 4. Materials and Methods

### 4.1. Laboratory Animals

The investigations were conducted on twelve immature (8–12 weeks old, 15–20 kg body weight, b.w.) female pigs of the Large White Polish breed. After the surgical and tracing procedures (see below), the animals were randomly divided into the control (C group; n = 6) and GUA-treated group (GUA group; n = 6). All the animals originated from a commercial fattening farm and were kept under standard laboratory conditions. They were fed standard fodder (Grower Plus, Wipasz, Wadag, Poland) and had free access to water. As the present study was designated to provide data concerning the chemical phenotypes of noradrenergic nerve fibers and CaMG neurons supplying the urinary bladder wall both under physiological and pathophysiological (GUA-treated animals) conditions, the authors decided to focus on sexually immature female animals in order to exclude any possible influences of reproductive hormones on studied tissues as identified in previous studies [126,127]. The animals were housed and treated in accordance with the rules of the Local Ethics Committee for Animal Experimentation in Olsztyn (affiliated to the National Ethics Commission for Animal Experimentation, Polish Ministry of Science and Higher Education; decision No. 94/2011 from 23 November 2011). All efforts were made to minimize the number of animals used and their suffering.

### 4.2. Anesthesia and Surgical Procedures

Before performing any surgical procedure, all animals were pretreated with atropine (Polfa, Lublin, Poland, 0.04 mg/kg b.w., s.c.) and azaperone (Stresnil, Janssen Pharmaceutica, Beerse, Belgium; 0.5 mg/kg b.w., i.m.), and after thirty minutes the main anesthetic drug, sodium pentobarbital (Tiopental, Sandoz, Poland; 0.5 g per animal, administered according to the effect), was given intravenously in a slow, fractionated infusion. The depth of anesthesia was monitored by testing the corneal reflex.

A mid-line laparotomy was performed in all the animals, and the urinary bladder was gently exposed to administer a total volume of 40 µL of 5% aqueous solution of the fluorescent retrograde tracer FB (Dr K. Illing KG & Co GmbH, Gross Umstadt, Germany) into the right side of the urinary bladder body wall in multiple injections (1 μL of the dye solution per 1 injection with a Hamilton microsyringe equipped with a 26S gauge needle) under the serosa along the whole extension of the urinary bladder dome keeping a similar distance between the places of injections. To avoid leakage of the dye, the needle was left in each place of FB injection for about one minute. The wall of the injected organ was then rinsed with physiological saline and gently wiped with gauze. Three weeks later, which is an optimal time for the retrograde tracer to be transported to the CaMG neurons [12,128], C group was treated with intravesical instillation of citrate buffer (pH 4.9; 60 mL per animal). The above-mentioned procedure was performed in the control pigs to ensure that changes in the chemical coding of noradrenergic nerve fibers and CaMG-UBPN after the GUA treatment were caused by this biological active substance itself, not due to the factors associated with the administration processes. The GUA group was treated with intravesical instillation of GUA (12 µg of GUA dissolved in 60 mL of citrate buffer, pH 4.9, per animal). Ten minutes after the infusion, the contents of the bladder were evacuated, and the catheter was removed. One week after the administration of citrate buffer or GUA, all the pigs were deeply anaesthetized (following the same procedure as describe above) and after the cessation of breathing, transcardially perfused with freshly prepared 4% paraformaldehyde in 0.1 M phosphate buffer (pH 7.4). Afterwards, the whole urinary bladder and bilateral CaMGs were collected from all animals. Tissue samples were then postfixed by immersion in the same fixative (10 min at room temperature), washed several times in 0.1 M phosphate buffer (pH 7.4; 4 °C; twice a day for three days) and finally transferred to and stored in 18% buffered sucrose at 4 °C (two weeks) until sectioning.

### 4.3. Sectioning of the Tissue Samples and Estimation of the Total Number of CaMG-UBPN

The urinary bladder wall samples analyzed in the present experiment were collected from the body of the bladder. To facilitate freezing, tissue cutting and analysis of cryostat sections, the corpus of the urinary bladder was first divided with scalpel into left and right part. Then each of these parts was divided into four (two upper and two lower) square blocks with side lengths of 1 cm by 1 cm. All the urinary bladder wall samples were cut with a HM525 Zeiss freezing microtome on transverse 12-µm-thick serial sections on chrome alum-gelatin-coated glass slides and subjected to immunohistochemical staining. Samples of the right as well as the left CaMG ganglia were cut with a HM525 Zeiss freezing microtome on transverse 10-µm-thick serial sections. Sections were put on chrome alum-gelatin-coated slides, air dried and examined under the fluorescent Olympus BX61 microscope equipped with a filter set specific for FB. To determine the relative number of bladder-projecting FB-positive neurons, cells were counted in every fourth section (to avoid double-counting of the same neuron; most neurons were approximately 40 μm in diameter) prepared from both the left and right ganglia of all animals. Only neurons with a clearly visible nucleus were considered. The results were pooled for every experimental animal, statistically analyzed and mean number of FB-positive cells was calculated. The statistical analysis was performed with Graph-Pad Prism 8 software (GraphPad Software, La Jolla, CA, USA). The total numbers of FB^+^ nerve cells counted in CaMGs and the relative frequencies of perikarya in the ganglia from either side were presented as mean ± standard deviation (S.D.)

### 4.4. Immunohistochemical Procedure

Double-immunofluorescence labelings were performed on cryostat sections prepared from each of the urinary bladder wall samples, as well as sections of right and left CaMG ganglia (selected from three different representative regions of the ganglion, located at upper one-third, middle and lower one-third, respectively), according to a previously described method [129]. The presence of all neurotransmitters (NPY, SOM, CB, VIP, GAL) or their markers (DβH, TH, nNOS; enzymes of the catecholamine or nitric oxide biosynthesis pathway, respectively), was previously revealed either in the intramural nerve fibers of the porcine urinary bladder [42,130] and/or in the CaMG-bladder projecting neurons of the pig [43,131]. It should be stressed that although both enzymes of the catecholamine synthesis pathway are synthesized in the cell body, DβH is very quickly transported from the cell body to the axon, which means that under physiological conditions, cell soma contains significantly fewer DβH molecules than TH, which makes it difficult to clearly determine under a light microscope, whether the studied ganglion cell belongs to the noradrenergic or non-adrenergic population. For this reason, in the present experiment we decided to use antibodies against TH as a tool for visualizing perikarya (the high concentration of the enzyme in the cell body (but not in its processes) allows a perfect distinction between noradrenergic and non-adrenergic cells), and anti-DβH antibodies as a tool for visualizing noradrenergic fibers in the sections of the bladder wall.

To determine the distribution and the chemical coding of noradrenergic intramural nerve fibers supplying the urinary bladder wall, a primary antiserum against DβH (marker of noradrenergic fibers) was applied in a mixture with antisera against: NPY, SOM, CB, VIP, GAL or nNOS, respectively. To analyze the immunohistochemical characteristics of CaMG bladder-projecting cells, primary antiserum toward TH (as a marker of catecholaminergic neurons; however, because it is commonly accepted that the main catecholamine utilized by the peripheral sympathetic neurons is NA, the term noradrenergic is applied throughout the paper) was applied in a mixture with antisera against NPY, SOM, CB, VIP, GAL or nNOS, respectively. The application of primary antisera raised in different species allowed to assess the coexistence of DβH or TH with other studied substances. Next, these primary antisera were visualized by secondary antisera conjugated to fluorescein isothiocyanate (FITC) or streptavidin-CY3 complex. Details concerning all the primary and secondary antibodies used in the present study are listed in Table 5.

The labelled sections were viewed under an Olympus BX61 microscope equipped with epifluorescence and an appropriate filter set for FB, CY3-conjugated streptavidin and FITC. Relationships between immunohistochemical staining and FB distribution were examined directly by interchanging filters. The images were taken with an Olympus XM10 digital camera (Tokyo, Japan). The microscope was equipped with cellSens Dimension 1.7 Image Processing software (Olympus Soft Imaging Solutions, Münster, Germany).

### 4.5. Estimation of the Density and Chemical Coding of Noradrenergic Nerve Fibers Supplying the Urinary Bladder and the CaMG-UBPN

The distribution and relative frequencies of labelled nerve fibers in the urinary bladder wall tissue samples were assessed semi-quantitatively [132,133] in 32 glass sections per one animal (4 fields (height 300 µm, length 400 µm) per section, 16 sections from both left and right side of the urinary bladder wall). The evaluation of these structures in the same preparations was performed independently by two investigators. The number of the nerve fibers immunoreactive to each substance was evaluated subjectively, based on a scale from—(when the nerve fibers were not found) to ++++ (a very dense meshwork of fibers). The exact scale of density assessment of nerve fibers in particular layers of the urinary bladder wall is presented in Table 6.

Sections containing FB^+^ neurons were chosen from representative regions (i.e., from the upper, middle and lower one-third of the ganglion) of each CaMG for immunohistochemical stainings. To determine the percentages of particular neuronal subpopulations, at least 300 of FB^+^ neuronal profiles were investigated with one combination of primary antisera and counted in both (left and right) CaMG in each animal studied. To avoid double counting of the same neurons, the neuronal cells were counted in every fourth section. The percentages of the retrogradely labelled neurons immunopositive to particular biologically active substances or their markers were pooled in all the animals and presented as mean ± SD. Morphometric data relative to each neuronal class were compared within each animal and among the animals and were analyzed by the Student’s *t* test using GraphPad PRISM 8.0 software. The differences were considered to be significant at *p* < 0.05.

### 4.6. Control of Specificity of the Tracer Staining and Immunohistochemical Procedures

Thorough macroscopic examinations of the sites of FB injections and the tissues adjacent to the urinary bladder were performed before the sample collection. The injection sites were easily identified by the yellow-labeled deposition left by the tracer within the bladder wall. Moreover, the sites of injection were also observed in the UV lamp rays in the dark room. The tissues adjacent to the bladder were not found to be contaminated with the tracer. To verify that the tracer had not migrated into the urethra, we analyzed, in cryostat sections and by means of the H&E staining techniques (IHC WORLD LLC, Woodstock, USA), possible signs of leakage of the tracer to the junction between the urinary bladder trigone and the cranial portion of the urethra. In all the animals studied no contamination of the urethra by the tracer was found. All these procedures excluded any leakage of the tracer and validated the specificity of the tracing protocol. To test the specificity of primary antibodies and staining reactions, preincubation tests were performed with the sections from the urinary bladder wall and CaMG ganglia of the control pigs (their inactivation by excess amount of antigens was performed before using to stain the control sections; Table 7).

Overnight preincubation of each primary antiserum at working dilution with 20 µg/mL of the respective antigen completely eliminated the immunoreaction. The omission and replacement of the respective primary antiserum with the corresponding non-immune serum also served as a negative control.

## 5. Conclusions

In conclusion, the present study for the first time demonstrates the ability of GUA to induce profound changes in the organization and chemical coding of nerve endings located in the wall of the porcine urinary bladder as well as at the level of their parent cell bodies in the CaMG, providing not only detailed data on the type of induced changes but also confirming the potential of GUA as a tool for studying adaptive changes in the peripheral nervous system.

## Figures and Tables

**Figure 1 ijms-22-04896-f001:**
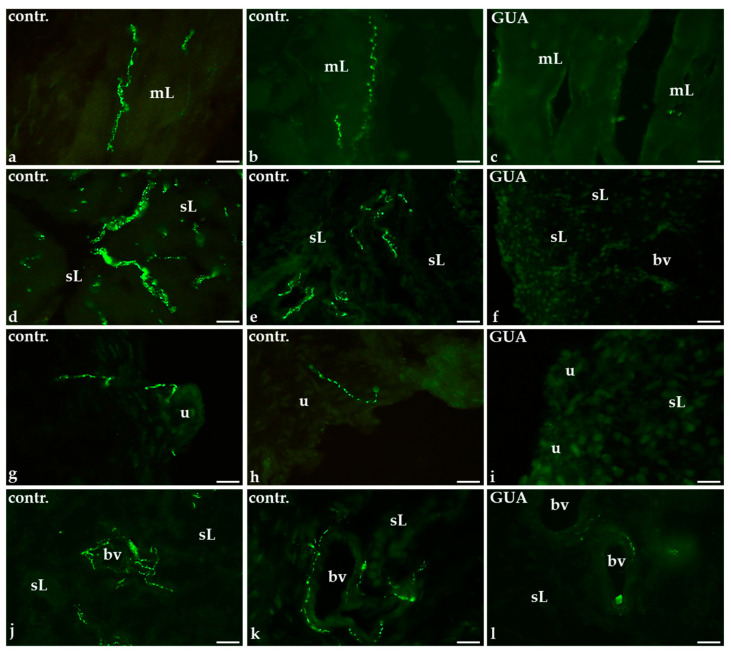
The distribution and relative frequency of dopamine β-hydroxylase (DβH)—immunoreactive (IR) nerve fibers in control (contr.) and guanethidine (GUA)—treated pigs; in the control animals few DβH-IR nerve fibers were observed in the muscle layer (mL; **a**,**b**); a moderate number of noradrenergic nerve terminals was present in the submucosal layer (sL; **d**,**e**); a single DβH-containing nerve fibers were found beneath the urothelium (u; **g**,**h**) and a very dense meshwork of noradrenergic nerve fibers supplied the wall of blood vessels (bv; **j**,**k**); after GUA treatment only a few nerve terminals were observed in the wall of blood vessels (bv; **l**) while no DβH-IR nerve fibers were found in the muscle layer (mL; **c**), submucosal layer (sL; **f**,**i**,**l**) or beneath the urothelium (u; **i**); ×20.

**Figure 2 ijms-22-04896-f002:**
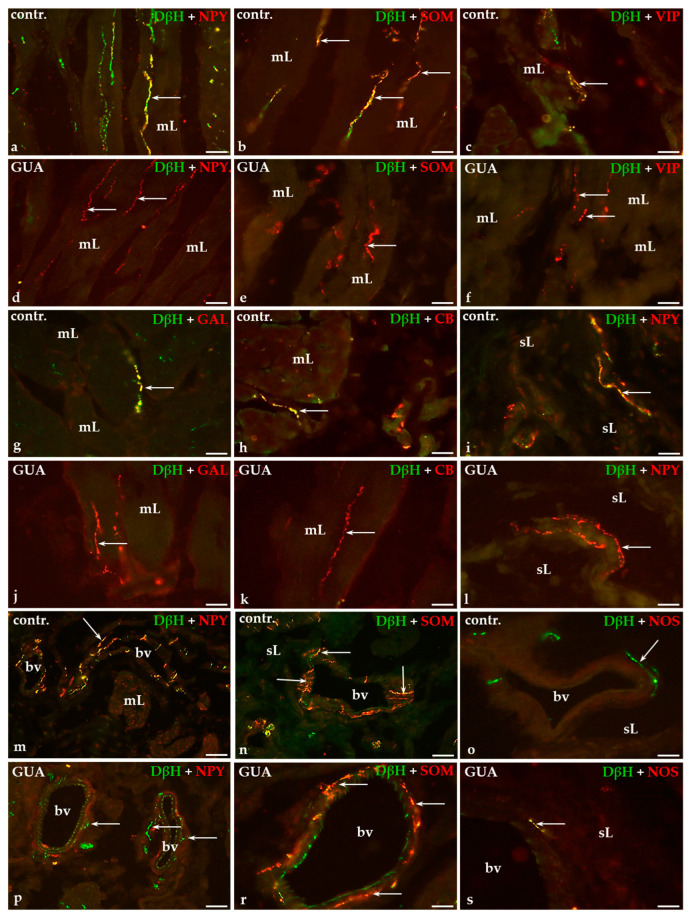
The distribution of DβH—(green labelled fibers) and neuropeptide Y (NPY)—(**a**,**d**,**i**,**l**,**m**,**p**), somatostatin (SOM)—(**b**,**e**,**n**,**r**), vasoactive interstitial polypeptide (VIP)—(**c**,**f**), galanin (GAL)—(**g**,**j**), calbindin (CB)—(**h**,**k**) or neuronal nitric oxide synthase (nNOS)-positive (**o**,**s**; red labelled fibers) nerve fibers in the muscle layer (mL), submucosal layer (sL) and blood vessels (bv) of the urinary bladder wall in the control (**a**–**c**,**g**–**i**,**m**–**o**) and GUA-treated (**d**–**f**,**j**–**l**,**p**–**s**) pigs. Red and green channels were digitally superimposed. Double-labelled fibers are yellow to orange and most of them are indicated with arrows; ×20 (**a**–**p**); ×40 (**r**,**s**).

**Figure 3 ijms-22-04896-f003:**
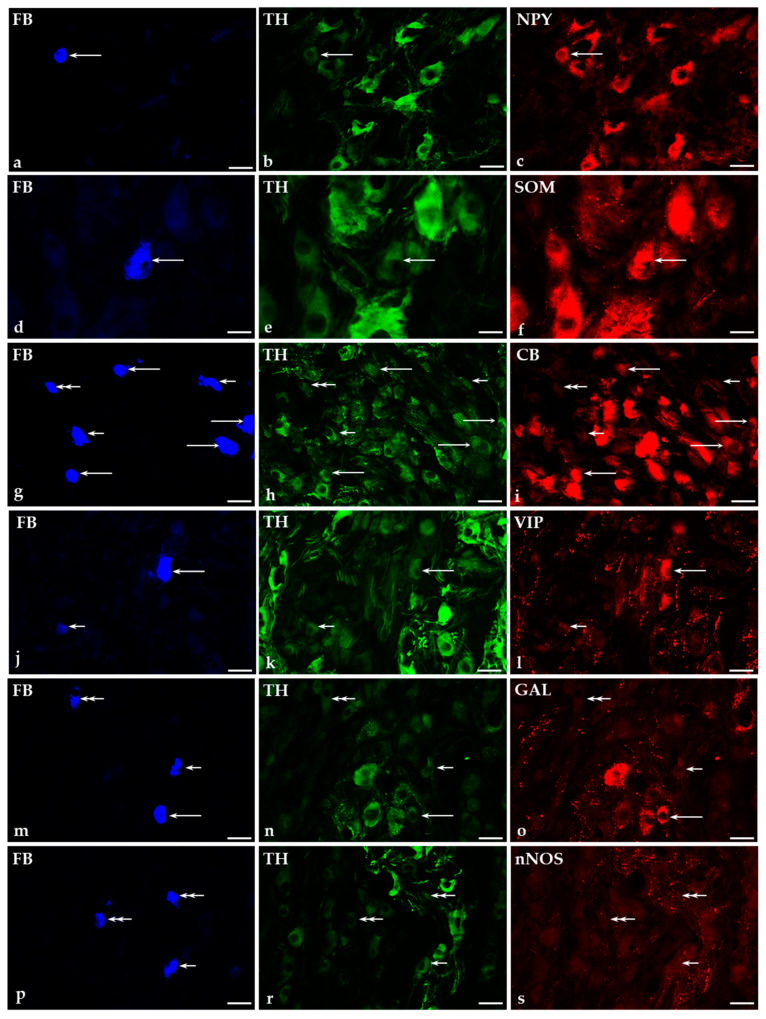
Representative images of caudal mesenteric ganglion-urinary bladder projecting neurons (CaMG-UBPN) in the control pigs. All the images were taken separately from blue (**a**,**d**,**g**,**j**,**m**,**p**), green (**b**,**e**,**h**,**k**,**n**,**r**) and red (**c**,**f**,**i**,**l**,**o**,**s**) fluorescent channels. Long arrows represent fast blue-positive (FB^+^) CaMG-UBPN (**a**,**d**,**g**,**j**,**m**), which were simultaneously tyrosine hydroxylase-positive (TH^+^; **b**,**e**,**h**,**k**,**n**) and NPY^+^ (**c**), SOM^+^ (**f**), CB^+^ (**i**), VIP^+^ (**l**) or GAL^+^ (**o**). Short arrows represent FB^+^ CaMG-UBPN (**g**,**j**,**m**,**p**), which were simultaneously TH^+^ (**h**,**k**,**n**,**r**) but CB-negative (CB^−^; **i**), VIP^−^ (**l**), GAL^−^ (**o**) or nNOS^−^ (**s**). Double arrows represent FB^+^ CaMG-UBPN (**g**,**m**,**p**), which were simultaneously both TH^−^ (**h**,**n**,**r**) and CB^−^ (**i**), GAL^−^ (**o**) or nNOS^−^ (**s**). Bar in images (**a**–**c**) and (**g**–**s**)—50 μm; Bar in images (**d**–**f**)—20 μm.

**Figure 4 ijms-22-04896-f004:**
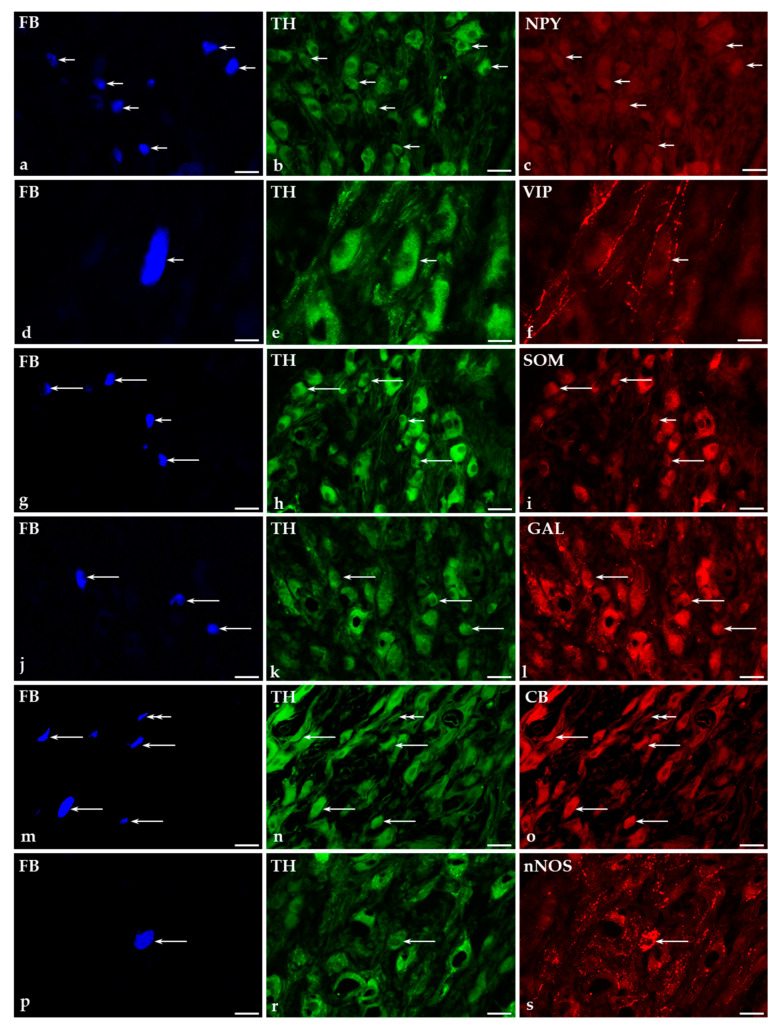
Representative images of CaMG-UBPN in the GUA-treated pigs. All the images were taken separately from blue (**a**,**d**,**g**,**j**,**m**,**p**), green (**b**,**e**,**h**,**k**,**n**,**r**) and red (**c**,**f**,**i**,**l**,**o**,**s**) fluorescent channels. Long arrows represent FB^+^ CaMG-UBPN (**g**,**j**,**m**,**p**), which were simultaneously TH^+^ (**h**,**k**,**n**,**r**) and SOM^+^ (**i**), GAL^+^ (**l**), CB^+^ (**o**) or nNOS^+^ (**s**). Short arrows represent FB^+^ CaMG-UBPN (**a**,**d**,**g**), which were simultaneously TH^+^ (**b**,**e**,**h**) but NPY^−^ (**c**), VIP^−^ (**f**) or SOM^−^ (**i**). Double arrows represent FB^+^ CaMG-UBPN (**m**), which were simultaneously both TH^−^ (**n**) and CB^−^ (**o**). Bar in images (**a**–**c**) and (**g**–**s**)—50 μm; Bar in images (**d**–**f**)—20 μm.

**Table 1 ijms-22-04896-t001:** The distribution and relative frequency of dopamine β-hydroxylase-immunoreactive (DβH-IR) nerve fibers supplying the porcine urinary bladder wall in the control group and guanethidine (GUA)group;—: lack of nerve fibers; +/−: single fibers; +: few fibers; ++: moderate number of fibers; +++: numerous fibers; ++++: a very dense meshwork of fibers; ↓: a decrease in the nerve fibers density.

Part of the Urinary Bladder Wall	Control Pigs	GUA-Treated Pigs
Muscle layer	+	−↓
Submucosal layer	++	−↓
Under the urothelium	+/−	−↓
Around blood vessels	++++	+↓

**Table 2 ijms-22-04896-t002:** The degree of colocalization of DβH and co-transmitters studied within the nerve fibers supplying the urinary bladder wall in control group and the GUA group. Legend: NPY: neuropeptide Y; SOM: somatostatin; VIP: vasoactive interstitial polypeptide; GAL: galanin; CB: calbindin; nNOS: neuronal nitric oxide synthase; mL: muscle layer; sL: submucosal layer; u: urothelium; bV: blood vessels;—: lack of nerve fibers; +/−: single fibers; +: few fibers; ++: moderate number of fibers; +++: numerous fibers; ++++: a very dense meshwork of fibers; ↓: a decrease in the nerve fibers density; ↑: an increase in the nerve fibers density.

Substances	Control Animals				GUA-Treated Animals			
	mL	sL	u	bv	mL	sL	u	bV
DβH/NPY	++++	+	+	++++	−↓	−↓	−↓	+/−↓
DβH/SOM	++	+	+	+/−	−↓	−↓	−↓	++++↑
DβH/VIP	+/−	−	−	−	−↓	−	−	−
DβH/GAL	+/−	−	−	−	−↓	−	−	−
DβH/CB	+/−	−	−	−	−↓	−	−	−
DβH/NOS	−	−	−	−	−	−	−	−/+↑

**Table 3 ijms-22-04896-t003:** Percentages of fast blue-positive (FB^+^) tyrosine hydroxylase containing (TH^+^) neuronal subpopulations in the caudal mesenteric ganglia (CaMG) of the control and GUA-treated animals, which simultaneously co-expressed neuropeptide Y (NPY^+^), somatostatin (SOM^+^), calbindin (CB^+^), vasoactive intestinal polypeptide (VIP^+^), galanin (GAL^+^) or neuronal nitric oxide synthase (nNOS^+^). Data expressed as mean ± SD. Asterisks mark statistically significant differences at *** *p* = 0.0001, **** *p* < 0.0001.

Experimental Groups	FB^+^/TH^+^ %	FB^+^/TH^+^/ NPY^+^ %	FB^+^/TH^+^/ SOM^+^ %	FB^+^/TH^+^/ CB^+^ %	FB^+^/TH^+^/ VIP^+^ %	FB^+^/TH^+^/ GAL^+^ %	FB^+^/TH^+^/ nNOS^+^ %
Control pigs	94.3 ± 1.8	89.6 ± 0.7	3.6 ± 0.4	2.06 ± 0.2	1.6 ± 0.2	1.6 ± 0.3	0
GUA-treated pigs	73.3 ± 1.4 ***	27.8 ±0.9 ****	68.7 ± 1.9 ****	9.1 ± 1.2 ***	0 ***	28.2 ± 1.3 ****	4.5 ± 0.6 ***

**Table 4 ijms-22-04896-t004:** Percentages of FB^+^ TH-immunonegative (TH^−^) neuronal subpopulations in the CaMG of the control and GUA-treated animals, which are simultaneously immunopositive to NPY^+^, SOM^+^, CB^+^, VIP^+^, GAL^+^ or nNOS^+^. Data are expressed as mean ± SD. Asterisks mark statistically significant differences at *** *p* = 0.0001, **** *p* < 0.0001.

Experimental Groups	FB^+^/TH^-^ %	FB^+^/TH^-^/ NPY^+^ %	FB+/TH^-^/ SOM^+^ %	FB^+^/TH^-^/ CB^+^ %	FB^+^/TH^-^/ VIP^+^ %	FB^+^/TH^-^/ GAL^+^ %	FB^+^/TH^-^/ nNOS^+^ %
Control pigs	5.7 ± 1.8	38.9 ± 6.7	0	0	14.8 ± 6.2	0	0
GUA-treated pigs	26.7 ± 1.4 ***	28.9 ± 0.8 ***	0	0	0 ****	0	0

**Table 5 ijms-22-04896-t005:** List of primary antisera and secondary reagents used in the study.

Antigen	Code	Dilution	Host	Supplier
**Primary Antibodies**				
DβH	MAB 308	1:1000	Mouse	Millipore; Temecula; CA; USA
TH	MAB318	1:800	Mouse	Millipore; Temecula; CA; USA
NPY	NA 1233	1:8000	Rabbit	Enzo Life Sciences; Farmingdale; NY; USA
SOM	T-4103	1:4000	Rabbit	Peninsula; San Carlos; CA; USA
CB	CB-38a	1:2000	Rabbit	Swant; Marly; Fribourg; Switzerland
VIP	VA 1285-0025	1:3000	Rabbit	Enzo Life Sciences; Farmingdale;NY; USA
GAL	AB 5909	1:2000	Rabbit	Millipore; Temecula; CA; USA
nNOS	AB5380	1:4000	Rabbit	Millipore; Temecula; CA; USA
**Secondary Reagents**				
Biotinylated anti-rabbit immunoglobulins	E 0432	1:1000	Goat	Dako; Hamburg; Germany
CY3-conjugated streptavidin	711-165-152	1:12,000	-	Jackson I.R.; West Grove; PA; USA
FITC-conjugated anti-mouse IgG	715-096-151	1:600	Donkey	Jackson I.R.; West Grove; PA; USA

**Table 6 ijms-22-04896-t006:** The scale used to estimate the density of nerve fibers in the urinary bladder wall.

Density Assessment Scale	Density Description
−	nerve fibers not found
+/−	single nerve fibers
+	few nerve fibers
++	moderate number of nerve fibers
+++	numerous nerve fibers
++++	a very dense meshwork of fibers

**Table 7 ijms-22-04896-t007:** List of antigens used in pre-absorption test.

Antigen	Code	Supplier
DβH	MBS9218238	MyBioSource; CA; USA
TH	AC21-0699-P	Abcore, Ramona, CA, USA
NPY	N3266	Sigma, St. Louis, MO, USA
SOM	S9129	Sigma, St. Louis, MO, USA
CB	AC21-2748-P	Abcore, Ramona, CA, USA
VIP	V6130	Sigma, St. Louis, MO, USA
GAL	G5773	Sigma, St. Louis, MO, USA
nNOS	N3033	Sigma, St. Louis, MO, USA

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
