# Peer review of "The Influence of an Adrenergic Antagonist Guanethidine on the Distribution Pattern and Chemical Coding of Caudal Mesenteric Ganglion Perikarya and Their Axons Supplying the Porcine Bladder"

_ijms, 2021, doi:10.3390/ijms22094896_

Round 1

Reviewer 1 Report

This paper by Bossowska et al. described the effects of guanethidine, an adrenergic antagonist, on the caudal mesenteric ganglion and nerve fibers in the urinary bladder. Data are comprehensive and pictures are impressive. However, there are several concerns which the authors should address before publication.

Major things

  1. Title: If possible, this reviewer recommends changing the title. Specialists may know the drug, guanethidine, however, non-specialists may not. Ordinal readers may cause a question what is guanethidine. The word “an adrenergic antagonist” is added, the reader can easily see this interesting paper. For example, the words “perikarya” and “wall” seem to be not essential.

  1. Abstract: It is not clear which part is about bladder wall and which part is about the caudal mesenteric ganglia. This reviewer recommends using the word “somata” or “cell bodies” in the part of caudal mesenteric ganglia.

  1. As markers of catecholaminergic nervous system, the authors used tyrosine hydoroxylase and dopamine b-hydroxylase. It is not clear why the authors used these two markers. The authors should clarify the reason why the authors utilized antibodies against two enzymes which associating with the formation of catecholamines.

  1. Discussion: There are several short paragraphs, in which a part of results is only repeated. This reviewer recommends shortening such paragraphs in one sentence, for example, and combining with the following paragraph.

Minor things

  1. P3, L112: The word “DbH-IR” is better than “noradrenergic nerve” because these staining profiles were not results of anti-noradrenaline antibody.

  1. P7, L214: Delete one space after “hydroxylase”.

  1. P12, L367: The word “noradrenaline” is better than “norepinephrine”.

  1. P13, L422: The words “TH+/NPY+ somata” is better than “TH+/NPY+ neurons”. And previous comma of this word is not necessary.

  1. Finally, in the manuscript this reviewer obtained, some words are painted in yellow. For example, “%” in line 246 and “=” in line 545, for example.

Author Response

Response to Reviewer 1 Comments

We wish to thank the Reviewer 1 for reading the manuscript and giving a positive
review. We appreciate all the detailed comments provided by the Reviewer which have helped
us to improve our contribution. The manuscript was moderated following the suggestions of
the Reviewer.
Below, please find our replies to the respective comments:

Major things

Point 1: Title: If possible, this reviewer recommends changing the title. Specialists may know
the drug, guanethidine, however, non-specialists may not. Ordinal readers may cause a question
what is guanethidine. The word “an adrenergic antagonist” is added, the reader can easily see
this interesting paper. For example, the words “perikarya” and “wall” seem to be not essential.
Response 1: In accordance with the recommendation of the Reviewer, the title of the
manuscript has been changed so that its content may be more accessible and understandable to
a wider group of readers and now reads as follows:
“The Influence of an Adrenergic Antagonist Guanethidine on the Distribution Pattern and
Chemical Coding of Caudal Mesenteric Ganglion Perikarya and Their Axons Supplying the
Porcine Bladder.”
Point 2: Abstract: It is not clear which part is about bladder wall and which part is about the
caudal mesenteric ganglia. This reviewer recommends using the word “somata” or “cell
bodies” in the part of caudal mesenteric ganglia.
Response 2: As recommended by the Reviewer, the abstract part describing the neurochemical
characteristics of the caudal mesenteric ganglion neurons was separated from the part
describing changes in the chemical coding of nerve fibers observed in the bladder wall by
adding the word “somata” or “cell bodies” as follows:
“Intravesical treatment with GUA led not only to a significant decrease in the number of bladder-projecting tyrosine hydroxylase (TH) CaMG somata (94.3 ± 1.8% vs. 73.3 ± 1.4%; control 
vs. GUA-treated pigs), but simultaneously resulted in the rearrangement of their cotransmitters repertoire, causing a distinct decrease in the number of TH+/NPY+ (89.6 ± 0.7%
vs. 27.8 ± 0.9%) cell bodies and an increase in the number of SOM- (3.6 ± 0.4% vs. 68.7 ±
1.9%), calbindin- (CB; 2.06 ± 0.2% vs. 9.1 ± 1.2%) or galanin-containing (GAL; 1.6 ± 0.3%
vs. 28.2 ± 1.3%) somata.
Point 3: As markers of catecholaminergic nervous system, the authors used tyrosine
hydroxylase and dopamine b-hydroxylase. It is not clear why the authors used these two
markers. The authors should clarify the reason why the authors utilized antibodies against two
enzymes which associating with the formation of catecholamines.
Response 3: We fully agree with the comments of the Reviewer that the text of the manuscript
should explain the use of two different adrenergic markers in this study. This explanation has
been included in the Materials and Methods section (line 621-630) and reads as follows:
“It should be stressed that although both enzymes of the catecholamine synthesis pathway are
synthesized in the cell body, DβH is very quickly transported from the cell body to the axon,
which means that under physiological conditions, cell soma contains significantly fewer DβH
molecules than TH, which makes it difficult to clearly determine under a light microscope,
whether the studied ganglion cell belongs to the noradrenergic or non-adrenergic population.
For this reason, in the present experiment we decided to use antibodies against TH as a tool for
visualizing perikarya (the high concentration of the enzyme in the cell body (but not in its
processes) allows a perfect distinction between noradrenergic and non-adrenergic cells), and
anti-DβH antibodies as a tool for visualizing noradrenergic fibers in the sections of the bladder
wall.”
Point 4: Discussion: There are several short paragraphs, in which a part of results is only
repeated. This reviewer recommends shortening such paragraphs in one sentence, for example,
and combining with the following paragraph.
Response 4: According to the suggestion of Reviewer some short paragraphs have been
combined with the following paragraph. It has been done as fallows:
- The sentence in line 421 was combined with sentence in line 420
- The sentence in line 462- 464 has been removed
- The sentence in line 468 was combined with sentence in line 467
- The sentence in line 491 was combined with sentence in line 490

Minor things

All minor corrections suggested by the Reviewer were introduced to the text of the
manuscript as follows:
Point 1: P3, L112: The word “DbH-IR” is better than “noradrenergic nerve” because these
staining profiles were not results of anti-noradrenaline antibody.
Response 1: corrected sentence in L112 - “A moderate number of DβH-IR terminals was
observed in the submucosal layer (Figure 1 d-e) and only single fibers penetrated beneath the
urothelium (Figure 1 g-h).”
Point 2: P7, L214: Delete one space after “hydroxylase”.
Response 2: corrected sentence in L215 - ”Under physiological conditions, both the tyrosine
hydroxylase-immunoreactive (TH-IR; 94.3 ± 2%; Figure 3b, e, h, k, n, r), as well as THimmunonegative (TH-; 5.7 ± 2%; Figure 3h, n, r) retrogradely-labeled bladder-projecting
neurons were found in both the ipsilateral and contralateral CaMGs studied.”
Point 3: P12, L367: The word “noradrenaline” is better than “norepinephrine”.
Response 3: corrected sentence in L369 - “On the other hand, it has been observed that
activation of α1 adrenergic receptors in the urothelium in rats [65], and the treatment of sensory
submucosal nerve fibers in mice [37] with phenylephrine leads to the release of some unknown
neurotransmitters from both sources, which in turn, facilitate detrusor relaxation, probably by
augmentation of noradrenaline release from the sympathetic nerves.”
Point 4: P13, L422: The words “TH+/NPY+ somata” is better than “TH+/NPY+ neurons”.
And previous comma of this word is not necessary.
Response 4: corrected sentence in L423 – “This finding is consistent with the significant
decrease in the number of urinary bladder-supplying TH+/NPY+ somata.”
Point 5: Finally, in the manuscript this reviewer obtained, some words are painted in yellow.
For example, “%” in line 246 and “=” in line 545, for example.
Response 5: In accordance with the Reviewer's instructions, all places marked in yellow in the
manuscript text have been corrected as follows:
Line 231 - the space between the numeric value and the percentage has been removed - (from
5.7 ± 2% in the control group to 26.7 ± 1% in GUA-treated animals)
Line 247 - the space between the numeric value and the percentage has been removed - (27.8
± 0.9%; Figure 4 a-c)
Line 262 - spaces on both sides of the equal sign have been added in the following equation -
p = 0.005
Line 289 - spaces on both sides of the equal sign have been added in the following equation -
p = 0.005
Line 546 - spaces on both sides of the equal sign have been added in the following equation –
n = 6
All new sentences or words added to the text are marked with red colour. We hope you
will find the revised version of the manuscript as suitable for publication in the International
Journal of Molecular Science.

With best wishes and kind regards,

Yours sincerely,
Agnieszka Bossowska

Reviewer 2 Report

Line 35: separate abovementioned

Line 56: write in full Th and L

Line 63: separate abovementioned

Line 82: add a reference after (NET)

Line 202: write in full FB

Line 211: compare statistically the results obtained from the distribution of FB neurons in control and GUA groups, or the results of the paragraphs 2.3 and 2.4

Line 291: add the after that

Line 545: explain the statistical account that led to this sample size

Line 553: explain why it was chosen to use only immature females and to exclude immature males

Line 575: delete the pigs (n=6) from and change were with was

Line 579-580: delete another six animals which served as and change were with was

Line 581: explain why it was chosen this amount of GUA

Line 587: delete studied

Line 590: add a time line where to insert the different treatments

Line 647: add the measures of the field

Line 649: add a dot after investigators and start a new sentence with The number. Delete bracket before number

Line 696: add a list of all abbreviations in the text

TABLE 1: write control group and GUA group instead of control pigs and GUA-treated pigs

TABLE 2: write control group and GUA group instead of control animals and GUA-treated animals

Figure 2: add figures regarding GAL and CB in GUA group. Reorganize all the panel. placing near the control and GUA images related to the same marker. Be careful to change the references in the text and in the figure legend

Author Response

We wish to thank the Reviewer 2 for reading the manuscript and giving a positive
review. We appreciate all the detailed comments provided by the Reviewer which have helped
us to improve our contribution. The manuscript was moderated following the suggestions of
the Reviewer.
Below, please find our replies to the respective comments:
Point 1: Line 35: separate abovementioned
Response 1: According to the suggestion of Reviewer the correction of sentence in line 36 has
been made as follows:
“These above mentioned functions are regulated by a complex neural control system containing
somatic, sympathetic, and parasympathetic components.”
Point 2: Line 56: write in full Th and L
Response 2: According to the Reviewer's recommendation, the abbreviations Th and L have
been explained with the full name in the sentence on line 56 as follows:
“Sympathetic pathway to the urinary bladder begins in the intermediolateral nuclei of the
thoracic (Th) and lumbar (L) segments of the spinal cord at the level Th10–L2 in humans and
L1–L5 in dog and cat [6].”
Point 3: Line 63: separate abovementioned
Response 3: According to the suggestion of Reviewer the correction of sentence in line 64 has
been made as follows:
“They reach the above mentioned ganglia via hypogastric nerve branches [16] and release
transmitters such as noradrenaline (NA) [17, 18], neuropeptide Y (NPY) [19], somatostatin
(SOM) [20] and galanin (GAL) [21].”
Point 4: Line 82: add a reference after (NET)
Response 4: The reference has been added after (NET) in sentence on line 82 as follows:
“GUA is transported across the sympathetic nerve membrane by special mechanism using
norepinephrine transporter (NET) [27].”
Point 5: Line 202: write in full FB
Response 5: It seems to us that it is not necessary to explain the full name of the FB+
abbreviation on line 202 of the manuscript text. According to the guidelines in the instructions
for authors is written “Abbreviations should be defined in parentheses the first time they appear
in the abstract, main text, and in figure or table captions and used consistently thereafter.”
The above mentioned abbreviation FB was first explained by its full name in the manuscript
text on line 181.
Point 6: Line 211: compare statistically the results obtained from the distribution of FB
neurons in control and GUA groups, or the results of the paragraphs 2.3 and 2.4
Response 6: In accordance with the Reviewer's recommendation, the results concerning the
comparison of the number and distribution of FB neurons between the control group and the
GUA group were placed in the text of the manuscript in line 211 as follows:
“The number and the distribution of UBPN in the GUA-treated animals in both the ipsilateral
and contralateral CaMG ganglia were similar to that observed in the control group.”
Point 7: Line 291: add the after that
Response 7: According to the suggestion of Reviewer the correction of sentence in line 293
has been made as follows:
“The results of the present study clearly indicate that the application of GUA is followed by
profound changes in the distribution, relative frequency, and chemical coding of sympathetic
nerve fibers as well as the noradrenergic CaMG neurons sup-plying the wall of the porcine
urinary bladder.”
Point 8: Line 545: explain the statistical account that led to this sample size
Response 8: The number of animals in the experimental groups is the smallest number that
allows for a satisfactory compromise between the statistical significance of the obtained results
and the regulations of the directives for the protection of laboratory animals (the 3R rule).
Point 9: Line 553: explain why it was chosen to use only immature females and to exclude
immature males
Response 9: Immature gilts were selected as a model for two reasons: i) sexual immaturity
allow to avoid any sexual cycle-related hormonal influences, eg., changes in the level of
circulating oestrogens, and hence, relevant receptor activation, which could lead to the
emergence of hormone-dependent rather than GUA-dependent changes in the chemical coding
of the studied cells), ii) the sex of the animals allowed for the reconstruction of the GUA
application route, accurately reflecting the methods of administration of this compound in
women.
Point 10: Line 575: delete the pigs (n=6) from and change were with was
Response 10: According to the suggestion of Reviewer the correction of sentence in line 577
has been made as follows:
“Three weeks later, which is an optimal time for the retrograde tracer to be transported to the
CaMG neurons [12, 129], C group was treated with intravesical instillation of citrate buffer
(pH 4.9; 60 ml per animal).”
Point 11: Line 579-580: delete another six animals which served as and change were with was
Response 11: According to the suggestion of Reviewer the correction of sentence in line 581-
582 has been made as follows:
“GUA group was treated with intravesical instillation of GUA (12 µg of GUA dissolved in 60
ml of citrate buffer, pH 4.9, per animal).”
Point 12: Line 581: explain why it was chosen this amount of GUA
Response 12: The amount of GUA administered to animals is the result of averaging the doses
used in previous studies, as reported in the literature, after conversion to µg/ml intravesical
infusion solution and adjustment to the weight of the animal.
Point 13: Line 587: delete studied
Response 13: According to the suggestion of Reviewer the correction of sentence in line 587-
588 has been made as follows:
“Afterwards, the whole urinary bladder and bilateral CaMGs were collected from all animals.”
Point 14: Line 590: add a time line where to insert the different treatments
Response 14: As suggested by the Reviewer, the time line of individual treatments has been
added to the text of the manuscript in the sentence on line 590-591 and reads as follows:
“Tissue samples were then postfixed by immersion in the same fixative (10 minutes at room
temperature), washed several times in 0.1 M phosphate buffer (pH 7.4; 4°C; twice a day for
three days) and finally transferred to and stored in 18% buffered sucrose at 4°C (two weeks)
until sectioning.”
Point 15: Line 647: add the measures of the field
Response 15: According to the suggestion of Reviewer the measures of the field has been
added in the sentence on line 657 as follows:
“The distribution and relative frequencies of labelled nerve fibers in the urinary bladder wall
tissue samples were assessed semi-quantitatively [133, 134] in 32 glass sections per one animal
(4 fields (height 300 µm, length 400 µm) per section, 16 sections from both left and right side
of the urinary bladder wall).”
Point 16: Line 649: add a dot after investigators and start a new sentence with The number.
Delete bracket before number
Response 16: According to the suggestion of Reviewer the correction of sentences in line 659-
662 has been made as follows:
“The evaluation of these structures in the same preparations was performed independently by
two investigators. The number of the nerve fibers immunoreactive to each substance was
evaluated subjectively, based on a scale from - (when the nerve fibers were not found) to ++++
(a very dense meshwork of fibers).”
Point 17: Line 696: add a list of all abbreviations in the text
Response 17: A list of all the abbreviations used in the manuscript text has been prepared and
placed on line 708.
Point 18: TABLE 1: write control group and GUA group instead of control pigs and GUAtreated pigs
Response 18: According to the suggestion of Reviewer the correction of sentence in the
TABLE 1 description has been made as follows:
“The distribution and relative frequency of dopamine β-hydroxylase-immunoreactive (DβHIR) nerve fibers supplying the porcine urinary bladder wall in control group and guanethidine
(GUA)-group;”
Point 19: TABLE 2: write control group and GUA group instead of control animals and GUAtreated animals
Response 19: According to the suggestion of Reviewer the correction of sentence in the
TABLE 2 description has been made as follows:
“The degree of colocalization of DβH and co-transmitters studied within the nerve fibers
supplying the urinary bladder wall in control group and GUA-group.”
Point 20: Figure 2: add figures regarding GAL and CB in GUA group. Reorganize all the
panel. placing near the control and GUA images related to the same marker. Be careful to
change the references in the text and in the figure legend
Response 20: In accordance with the Reviewer's recommendation, figures regarding GAL and
CB in GUA group has been added in Figure 2. Additionally, the reorganization of all the panel
has been made and all GUA images have been placed near the control images related to the 
same marker. Furthermore, the references in the text and in the Figure 2 legend have been
changed.
All new sentences or words added to the text are marked with red colour. We hope you
will find the revised version of the manuscript as suitable for publication in the International
Journal of Molecular Science.
With best wishes and kind regards,
Yours sincerely,
Agnieszka Bossowska